# Decreased FXR Agonism in the Bile Acid Pool Is Associated with Impaired FXR Signaling in a Pig Model of Pediatric NAFLD

**DOI:** 10.3390/biomedicines11123303

**Published:** 2023-12-13

**Authors:** Magdalena A. Maj, Douglas G. Burrin, Rodrigo Manjarín

**Affiliations:** 1Department of Biological Sciences, California Polytechnic State University, San Luis Obispo, CA 93407, USA; 2Center for Applications in Biotechnology, California Polytechnic State University, San Luis Obispo, CA 93407, USA; 3USDA-ARS Children’s Nutrition Research Center, Department of Pediatrics, Baylor College of Medicine, Houston, TX 77030, USA; doug.burrin@usda.gov; 4Department of Animal Science, California Polytechnic State University, San Luis Obispo, CA 93407, USA; rmanjari@calpoly.edu

**Keywords:** transcriptomics, liver, Iberian pig, metabolomics

## Abstract

The objective of this study was to investigate whether the impairment of farnesoid X receptor (FXR)-fibroblast growth factor 19 (FGF19) signaling in juvenile pigs with non-alcoholic fatty liver disease (NAFLD) is associated with changes in the composition of the enterohepatic bile acid pool. Eighteen 15-day-old Iberian pigs, pair-housed in pens, were allocated to receive either a control (CON) or high-fructose, high-fat (HFF) diet. Animals were euthanized in week 10, and liver, blood, and distal ileum (DI) samples were collected. HFF-fed pigs developed NAFLD and had decreased FGF19 expression in the DI and lower FGF19 levels in the blood. Compared with the CON, the HFF diet increased the total cholic acid (CA) and the CA to chenodeoxycholic acid (CDCA) ratio in the liver, DI, and blood. CA and CDCA levels in the DI were negatively and positively correlated with ileal FGF19 expression, respectively, and blood levels of FGF19 decreased with an increasing ileal CA to CDCA ratio. Compared with the CON, the HFF diet increased the gene expression of hepatic 12-alpha-hydrolase, which catalyzes the synthesis of CA in the liver. Since CA species are weaker FXR ligands than CDCA, our results suggest that impairment of FXR-FGF19 signaling in NAFLD pigs is associated with a decrease in FXR agonism in the bile acid pool.

## 1. Introduction

The prevalence of non-alcoholic fatty liver disease (NAFLD) in children has increased from 36/100,000 in 2009 to 58.2/100,000 in 2018 [1] and currently represents the main cause of pediatric chronic liver disease in the United States [2,3]. Overnutrition is a key factor that affects the progression and development of NAFLD, with the majority of children with NAFLD being overweight or obese [2,4]; nonetheless, NAFLD is also diagnosed in children with a normal body mass index [5]. Pediatric NAFLD represents a spectrum of disease ranging from hepatic steatosis to non-alcoholic steatohepatitis (NASH), which is associated with inflammation and cell injury [6]. NASH can revert with intervention but can also progress to cirrhosis [7] and is associated with a higher risk of developing hepatocellular carcinoma [8,9]. There are currently no approved medications for the treatment of NAFLD in children, and the only management strategy involves lifestyle changes to reduce total ingested energy and an increase in physical activity [10].

One molecular mechanism implicated in the pathogenesis of NAFLD is an impairment of the function of the farnesoid X receptor (FXR) [11], which is primarily activated by bile acids (BAs) in the liver and gut (Figure 1).

Upon binding BAs to the hepatic FXR, the FXR-retinoid X receptor (RXR) heterodimer complex is activated, resulting in the induction of the transcriptional repressor small heterodimer partner (SHP) [11]. The SHP downregulates sterol-regulatory element-binding protein 1c, limiting hepatic de novo lipogenesis [11]. In addition, FXR signaling promotes β-oxidation by activating the peroxisome-proliferator-activated receptor and repressing key enzymes necessary for gluconeogenesis, reducing hepatic glucose output [11]. Transcriptional events exerted by the FXR-SHP in the liver decrease BA synthesis by downregulating the hepatic enzymes sterol 7-alpha-hydrolase (CYP7A1) and sterol 12-alpha-hydroxylase (CYP8B1) [12,13]. The FXR is also activated by BAs in the ileum, where it induces the expression of the intestinal hormone fibroblast growth factor 19 (FGF19) [14]. FGF19 travels to the liver with the portal blood and activates a hepatic cell surface receptor, FGF receptor 4 (FGFR4), and its co-receptor, β-klotho, which initiates a Src-mediated phosphorylation cascade and promotes the activation of the FXR in hepatocytes [14] (Figure 1). As a result of these effects, enterohepatic FXR-FGF19 signaling protects against steatosis and the cytotoxic accumulation of BAs in the liver. Besides its role in lipid and BA metabolism, FXR signaling also promotes protein catabolism and ammonium clearance by binding directly to genes of amino acid degradation and ureagenesis [15]. In addition, activation of the FXR shows anti-inflammatory properties by repressing hepatocellular NF-kappa B activation and reducing pro-inflammatory cytokines such as tumor necrosis factor α (TNFα) and iNOS in the liver [16].

Many studies point toward an impairment of FXR-FGF19 signaling as a key factor in the pathogenesis of pediatric NAFLD. Children with NAFLD exhibit lower fasting and postprandial FGF19 levels in the blood and have significantly higher BA levels and hepatic expression of CYP7A1 than those without NAFLD, suggesting that the FXR may not negatively regulate CYP7A1 directly or through FGF19 signaling [17,18,19,20]. The etiology of impaired FXR activation in patients with NAFLD has not been elucidated; however, it may be caused by an unfavorable BA composition. BAs differ in their potency to activate the FXR, in the order chenodeoxycholic acid (CDCA) > deoxycholic acid (DCA) > cholic acid (CA) [21]. Interestingly, in adults with NAFLD, the median ratio of total cholate (CA + gCA + tCA) to total chenodeoxycholate (CDCA + tCDCA + gCDCA) in plasma increased with the severity of the disease [22,23]. Similarly, the percent quantity of serum DCA, which is derived from CA by microbial activity in the gut [24], was higher than CDCA in children with NAFLD [17]. CA differs from CDCA in that it has an additional hydroxyl on carbon 12 of the steroidal scaffold installed by the enzyme CYP8B1 of the classic BA pathway [13] (Figure 1). As a result, CYP8B1 regulates the synthesis of CA and the overall CA:CDCA ratio of the bile pool [13]. In this regard, CYP8B1 knockdown in rodent models of NAFLD significantly lowered hepatic lipid content [25,26], which suggests that a decrease in CYP8B1 expression may protect against NAFLD. Moreover, reduced CYP8B1 activity in patients with CYP8B1 loss-of-function mutations was associated with decreased insulin resistance, which is a risk factor for NAFLD [27]. Despite this preliminary evidence, it remains unknown whether changes in CYP8B1 expression and CA:CDCA ratio impact enterohepatic FXR signaling and NAFLD progression.

We have previously established a pig model of pediatric NAFLD in which juvenile Iberian pigs fed a liquid diet enriched in sugars and fats for 10 weeks developed NASH, cholestasis, and impairment of the enterohepatic FXR-FGF19 pathway [28,29,30]. Therefore, the objective of this study was to investigate whether the impairment of FXR-FGF19 signaling in diet-induced NAFLD pigs is associated with the upregulation of CYP8B1 expression in the liver and an increased enterohepatic CA to CDCA ratio.

## 2. Materials and Methods

### 2.1. Animals and Experimental Design

These studies were approved by the Institutional Animal Care and Use Committee of California Polytechnic State University (#1611) and followed the guidelines issued by the National Research Council Guide for the Care and Use of Laboratory Animals. Eighteen male (M) and female (F) juvenile Iberian pigs (15 ± 3 d of age) were housed in pairs and randomly allocated to receive for 10 weeks 1 of 2 liquid diets (g/kg body weight (BW)/d; Table 1 and Table 2): (1) control (CON, n = 8 pigs, 6M/2F): 0 g fructose, 11.2 g fat, and 199.3 kcal metabolizable energy (ME), and (2) high-fat high-fructose (HFF, n = 10 pigs, 6M/4F): 21.6 g fructose, 16.7 g fat, and 302.6 kcal ME. The utilization of a high-fructose and high-fat diet is a standard way to produce fatty liver in animal models of the disease [31].

The composition of the CON and HFF diets was based on our previous work [28,30]. Animals were fed 45 mL/kg body weight (BW) at 6-h intervals to match the physiological volume of food consumed by pigs. BW was recorded every 3 d, and food intake was adjusted accordingly. Pigs were euthanized on week 10 at 8 h post-feeding, as previously described [28,30], and tissue from the left medial segment of the liver and the distal section of the ileum (DI) was collected immediately after euthanasia.

### 2.2. Serum Biochemistry and Hormones

Blood was collected from the left ventricle immediately before euthanasia. Blood samples were centrifuged at 2100 RCF for 15 min at 4 °C, and both serum and plasma were stored at −80 °C to be used in further analyses. Serum FGF19 levels and lipid and liver biochemistries were measured as previously described [28,30].

### 2.3. Histology and Immunohistochemistry

Liver tissue stains were processed as previously described [28]. Stains were semi-quantitatively evaluated for steatosis, ballooning, degeneration, Mallory-denk bodies, lobular inflammation, fibrosis, and necrosis by a pathologist blinded to the treatment groups. To assess the cellular proliferation, sections were incubated with an anti-Ki67 antibody, and the number of Ki67^+^ hepatocytes was quantified using the multi-point tool in ImageJ software bundled with Java 8 [32]. Triacylglycerides (TAGs) were extracted from whole liver homogenates and quantified following previously described methods [28,30].

### 2.4. Metabolomics and Transcriptomics Analyses

Absolute levels of BAs in the liver, plasma, and DI were quantified by protein precipitation extraction with ultra-performance liquid chromatography-tandem mass spectrometry as previously described [28,30]. Values were expressed as peak areas under the curve. Transcriptomics of the liver and DI were performed by GENEWIZ, LLC. (South Plainfield, NJ, USA) following the methods previously described [30].

### 2.5. Statistical Methods

Univariate parameters were analyzed by a one-way ANOVA using a mixed model in SAS 9.2 (PROC MIXED, SAS Institute Inc., Cary, NC, USA) that included diet as a fixed effect and pen nested in diet as a random effect. The data are presented as means ± SD. Multiple comparisons were corrected with the Tukey post hoc test, and significant effects were considered at *p* ≤ 0.05. Liver scoring was analyzed by the Kruskal–Wallis test with the Bonferroni multiple comparisons test for non-parametric data (PROC NPAR1WAY and PROC RANK, SAS). Metabolomic data was analyzed using the %polynova_1way SAS macro [33]. Transcriptomic data were analyzed using a generalized linear model with an assumption of a negative binomial distribution of gene counts and using a 5% FDR threshold in edgeR—Bioconductor packageBio package,, as previously described [34]. The Database for Annotation, Visualization, and Integrated Discovery (DAVID) software version 6.8 [35] was used to perform functional enrichment analyses on differently expressed genes.

## 3. Results

HFF-fed pigs developed hepatomegaly, steatosis, and increased hepatocellular proliferation (*p* ≤ 0.01; Figure 2A).

Hepatic steatosis was classified as macro- and microvesicular, with periportal or diffuse distribution. In addition, 30% of HFF-fed pigs showed histopathological lesions consistent with NASH, including hepatocellular ballooning, Mallory hyaline, and lobular inflammation (*p* ≤ 0.001; Figure 2B,C). None of the pigs developed fibrosis, likely due to the short duration of the study. Consistent with histopathological changes, serum biochemistry also showed an increase in markers of liver injury, including alanine transaminase (*p* ≤ 0.05), aspartate transaminase (*p* ≤ 0.01), gamma-glutamyl transferase (*p* ≤ 0.05), lactate dehydrogenase (*p* ≤ 0.01), and total bilirubin (*p* ≤ 0.05) in HFF-fed pigs compared with the CON (Table 3).

Conversely, glucose (*p* ≤ 0.01), creatinine (*p* ≤ 0.05), blood urea nitrogen (*p* ≤ 0.001), cholesterol (*p* ≤ 0.05), and high- and low-density lipoproteins (*p* ≤ 0.05) decreased in the HFF group compared with the CON group (*p* ≤ 0.05, Table 3). No differences were found in the lean mass composition or BW gain between the CON and HFF groups (Figure 3A,B).

The liver injury in HFF-fed pigs was further characterized through transcriptome-wide RNA profiling, which showed the upregulation of biological processes associated with the immune and inflammatory response (*p* ≤ 0.01, Table 4).

Upregulated genes associated with an immune response in the liver of the HFF pigs included toll-like receptors 4, 8, and 9, interleukins B1 and 8, several chemokine and TNF receptors, and the class II major histocompatibility complex (Figure 4).

In addition, the transcriptional profiles highlighted the downregulation of genes in the HFF group associated with mitochondrial function and lipid metabolism, including fatty-acid beta oxidation, the urea cycle, and one-carbon metabolism (*p* ≤ 0.01, Table 4 and Figure 4). On a pathway level, DAVID analysis of the DI transcriptome identified HFF-induced upregulation of biological processes associated with cholesterol, sphingolipid and medium-chain fatty acid metabolism, fatty acid oxidation, and TAG catabolism (*p* ≤ 0.05, Table 5 and Figure 5).

Upregulated genes associated with lipid metabolism in the DI of the HFF pigs included carnitine palmitoyltransferase I, carnitine O-acetyltransferase, apolipoprotein C-III, lipoprotein lipases, and a patatin-like phospholipase domain containing 1 and apolipoprotein C3 (Figure 5).

We next analyzed plasma levels of FGF19 and the expression of FGF19 in the DI. Circulating FGF19 decreased in the HFF group compared with the CON (*p* ≤ 0.001), whereas transcriptomic analysis of the DI showed a decreasing trend in FGF19 expression (*p* ≤ 0.1) for the HFF-fed pigs compared with the CON (Figure 6A).

To investigate whether changes in FGF19 were associated with decreased FXR agonism in the BA pool, we quantified CA and CDCA species in the liver, DI, and blood and correlated them with ileal gene expression and blood levels of FGF19. Data are presented as percent quantities of unconjugated and conjugated CA (i.e., CA + tCA + gCA) and CDCA (i.e., CDCA + tCDCA + gCDCA) species in the BA pool. Compared with the CON, the HFF diet increased the total CA and CA to CDCA ratio in the liver, DI, and blood (*p* ≤ 0.05; Figure 6B). Moreover, total CA and CDCA in the DI were negatively (R^2^ = 0.25, *p* ≤ 0.05) and positively (R^2^ = 0.34, *p* ≤ 0.01) correlated, respectively, with ileal FGF19 expression (Figure 6C). In addition, circulating FGF19 decreased with increasing ileal CA:CDCA ratio (R^2^ = 0.57, *p* ≤ 0.01; Figure 6C). Finally, we analyzed the expression of liver enzymes involved in BA synthesis. Compared with the CON, HFF-fed pigs showed upregulation of CYP8B1 (*p* ≤ 0.05; Figure 6D), while sterol 7-alpha-hydroxylase (CYP7B1), which catalyzes the last step in the synthesis of CDCA (Figure 1), was lower in the HFF group compared to the CON. There were no differences in the expression of CYP7A1, CYP27A1, and CYP4A21.

## 4. Discussion

The objective of this study was to investigate whether a decrease in FXR signaling in diet-induced NAFLD was associated with changes in BA composition in a pig model of NAFLD by using a multiomics approach combined with histology and serum biochemistry. After 10 weeks of HFF feeding, juvenile Iberian pigs developed NAFLD, had decreased levels of FGF19 in the serum, decreased FGF19 expression in the DI, and increased levels of total CA and CA to CDCA ratio in the liver, DI, and blood. In addition, HFF-fed pigs had increased expression of the hepatic CYP8B1 enzyme, which catalyzes the synthesis of CA. Transcriptomic analysis of liver tissue also showed the downregulation of multiple genes associated with beta-oxidation and the urea cycle in the HFF group compared with the CON, whereas expression of pro-inflammatory pathways such as NF-kappa B and tumor necrosis factor α was increased in HFF-fed pigs. Taken together, our results indicate an enterohepatic impairment of FXR signaling in NAFLD pigs, with dysregulation of FXR downstream metabolic and inflammatory pathways in the liver and reduced FXR-induced FGF19 synthesis in the DI. Of note, expression of DI genes involved in fatty acid beta oxidation, lipid transport, and TAG catabolism was upregulated in HFF-fed pigs compared with the CON, suggesting that lipid metabolism was not decreased in the gut mucosa of NAFLD pigs.

The dysregulation of hepatic FXR signaling in pediatric NAFLD is well established. Blood levels of FGF19 are decreased in children with NAFLD [17,18,19,20], whereas circulating BAs and hepatic expression of CYP7A1 are higher than in those without NAFLD [17]. Moreover, fasting plasma 7α-hydroxy-4-cholesten-3-one (C4), a marker used to monitor CYP7A1 enzymatic activity [36], is elevated in NAFLD patients, supporting the notion of increased BA synthesis due to lower FXR activity [22,37]. Previous studies suggest that decreased FXR activation may reflect an obesity and insulin resistance (IR)-mediated phenomenon. In this regard, increased plasma BA concentrations are observed in patients with type-2 diabetes, IR, and obesity [38,39,40], and both obesity and IR are considered key factors in the pathogenesis of pediatric NAFLD [41]. Rodent models of type-2 diabetes also exhibit reduced hepatic FXR expression, paralleled by an increased expression of CYP7A1 in the liver and an increased BA pool [42]. Moreover, physiological levels of insulin downregulate CYP7A1 gene transcription in cultured rat hepatocytes, with a consequent suppression of BA synthesis [43]. However, juvenile Iberian pigs with NAFLD had decreased FGF19 expression in the absence of obesity and IR, suggesting that FXR dysregulation cannot be attributed solely to underlying metabolic conditions. In children with NAFLD, gene expression of β-Klotho [20], a hepatic coreceptor for FGF19, and FXR protein abundance [19] are decreased in liver tissue, and their levels are inversely correlated with disease severity. Interestingly, gene expression of the FXR and the FGF19 receptor in liver tissue was not altered in NAFLD pigs, suggesting that the impairment of FXR signaling in NAFLD pigs may be associated with reduced FXR activation. Since CA species are weaker agonists than CDCA [21], it is plausible that the increase in the CA to CDCA ratio in HFF-fed pigs reduced FXR activation and thus impaired intestinally derived FGF19 production and signaling in the liver.

The development of NAFLD in the absence of obesity and insulin resistance in the HFF-fed pigs was expected, as we have previously shown that our pediatric pig model resembles choline-deficient dietary models of NASH [28,29,30], in which animals develop liver injury in the absence of other metabolic abnormalities [44,45]. The lack of obesity is possibly due to increased energy expenditure caused by the development of a hypermetabolic condition in the juvenile pigs [28,46]. It is important to emphasize that our HFF diet was not choline or methionine deficient. We speculate that the choline-deficient phenotype is induced by feeding a high-fat diet, which may deplete choline levels in the liver to synthesize bile to absorb the excess fat [28,30]. A depletion in choline levels can also explain the downregulation of one-carbon metabolism genes in the liver and the decrease in serum levels of cholesterol and low- and high-density lipoproteins in the HFF group compared to the CON. Decreased choline availability reduces the hepatic synthesis of PCs by the cytidine 5’-diphosphocholine pathway, which then impairs the production of very low-density lipoproteins and the export of fat and cholesterol from the liver [47].

Bile acids are synthesized from cholesterol by alternative and classical pathways (Figure 1). The alternative pathway is catalyzed by CYP27A1 and CYP7B1 and predominantly produces CDCA [13], which in pigs is further converted into hyocholic acid by the enzyme CYP4A21 [48]. The classical pathway is initiated by 7α-hydroxylation of cholesterol by CYP7A1 to form C4, followed by 12α-hydroxylation by CYP8B1 to produce CA [13]. However, without 12α-hydroxylation, C4 is converted to CDCA by CYP27A1 [49]. As a result, CYP8B1 is the only enzyme catalyzing the synthesis of CA. Research indicates that knocking out CYP8B1 in mouse models of NAFLD improves hepatic steatosis [25,26]. Similarly, downregulation of CYP8B1 expression after bariatric surgery in a high-fat diet-induced obese mouse model was associated with lower hepatic TAG content [50], which suggests that a decrease in CYP8B1 activity may protect against NAFLD. Our findings support these previous studies in that NAFLD in HFF-fed pigs was associated with increased hepatic expression of CYP8B1. The enzyme CYP8B1 is regulated by a negative feed-back loop by BA-activation of the FXR in the liver, which in turn reduces the proportion of CA in bile [13] (Figure 1). Accordingly, Xu et al. [51] showed that FXR-agonist obeticholic acid decreased CYP8B1 expression and reduced CA levels by ∼71% in a mouse model of atherosclerosis. Similarly, oral administration of the FXR agonist tropifexor decreased CYP8B1 in liver tissue from rats [52]. Conversely, expression of CYP8B1 is upregulated by PPARα [53], which is a ligand-activated transcription factor that is activated by fatty acids and their derivatives. We speculate that dietary fat in HFF-fed pigs upregulated CYP81B and promoted changes in the composition of the BA pool to be less FXR agonistic, which may have blunted ileal FGF19 synthesis, thus decreasing the protective effects of the FXR against intrahepatic fat. In support of this hypothesis, Bisschop et al. [54] showed that a high-fat, low-carbohydrate diet increased the molar ratio of CA and decreased CDCA in the blood when compared with a low-fat, high-carbohydrate diet. Similarly, a high-fat diet increased the hepatic CA to CDCA ratio in obese rats [55]. Our previous studies also suggest that high-fat diets alter the composition of the BA pool to be less FXR-agonistic. Juvenile Iberian pigs fed hypercaloric diets formulated to have low normal levels of fat, so the excess of calories would be provided solely by sugars, had levels of systemic BAs within an expected physiological range and a small decrease in circulating FGF19 [56]. Conversely, isocalorically, substitution of dietary lard with coconut oil in a high-fat, high-fructose diet increased primary BAs, caused a 10-fold decrease in circulating FGF19, and promoted a greater degree of hepatic injury in juvenile Iberian pigs [30]. Taken collectively, these studies suggest that dietary fat has a greater impact on BA composition and FGF19 dysregulation than carbohydrates.

## 5. Conclusions

In conclusion, decreased FGF19 in a pig model of pediatric NAFLD was associated with increased CYP8B1 expression in the liver and an increased CA to CDCA ratio in the enterohepatic BA pool. Future studies are needed to elucidate whether dietary fat regulates CYP8B1 and promotes NAFLD and to apply this knowledge to establish novel mechanism-based interventions to treat the disease.

## Figures and Tables

**Figure 1 biomedicines-11-03303-f001:**
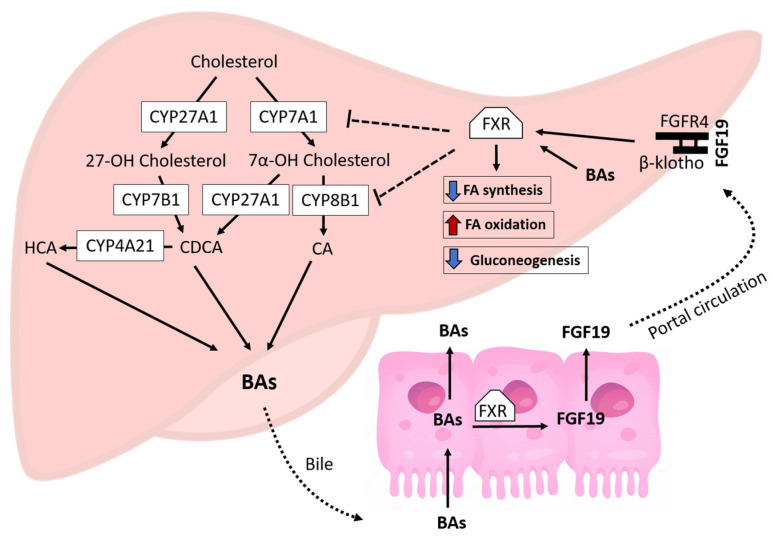
Enterohepatic regulation of bile acids, lipids, and glucose metabolism by nuclear Farnesoid-X receptor (FXR)-fibroblast growth factor 19 (FGF19) signaling. BAs, bile acids; CA, cholic acid; CDCA, chenodeoxycholic acid; CYP27A1, sterol 27-hydroxylase; CYP7A1, sterol 7-alpha-hydroxylase; CYP7B1, oxysterol-7-alpha-hydroxylase; CYP8B1, sterol 12-alpha-hydroxylase; CYP4A21, taurochenodeoxycholic acid 6-alpha-hydroxylase; FGFR4, fibroblast growth factor receptor 4; HCA, hyocholic acid.

**Figure 2 biomedicines-11-03303-f002:**
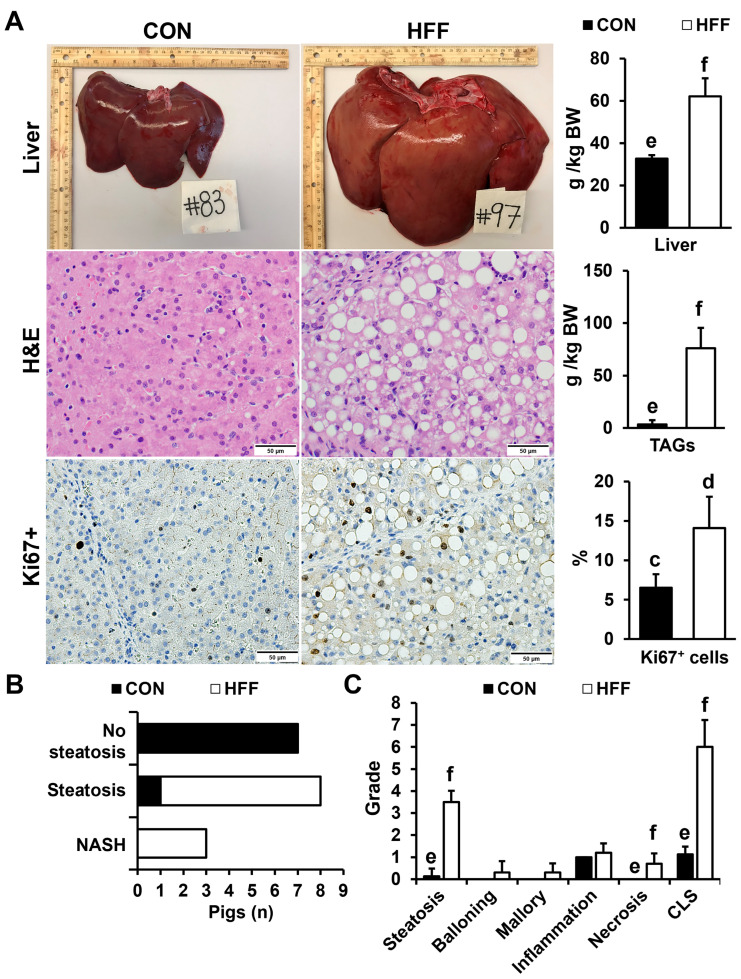
(**A**) Representative livers and liver stains from pigs fed control (CON) and high-fat, high-fructose (HFF) diets, taken immediately after euthanasia on day 70 of the study. Pictures were captured with a final magnification of 40× using an optical microscope (Axioscop 40; Zeiss, Thornwood, NY, USA) fitted with a camera (AxioCam MRc; Zeiss). The HFF diet increased relative organ weight, triacylglyceride content (TAG), and the percentage of Ki67+ cells, representative of active cell division in the liver. (**B**) The HFF diet increased the number of pigs with steatosis and non-alcoholic steatohepatitis (NASH) compared with the CON. (**C**) The grade of steatosis, necrosis, and composite liver score (CLS) increased in the HFF group compared with the CON. Values are means ± SD. *p*-values were adjusted for multiple testing with Tukey’s post hoc test. Letters represent significant differences between groups. ^cd^
*p* ≤ 0.01, ^ef^
*p* ≤ 0.001.

**Figure 3 biomedicines-11-03303-f003:**
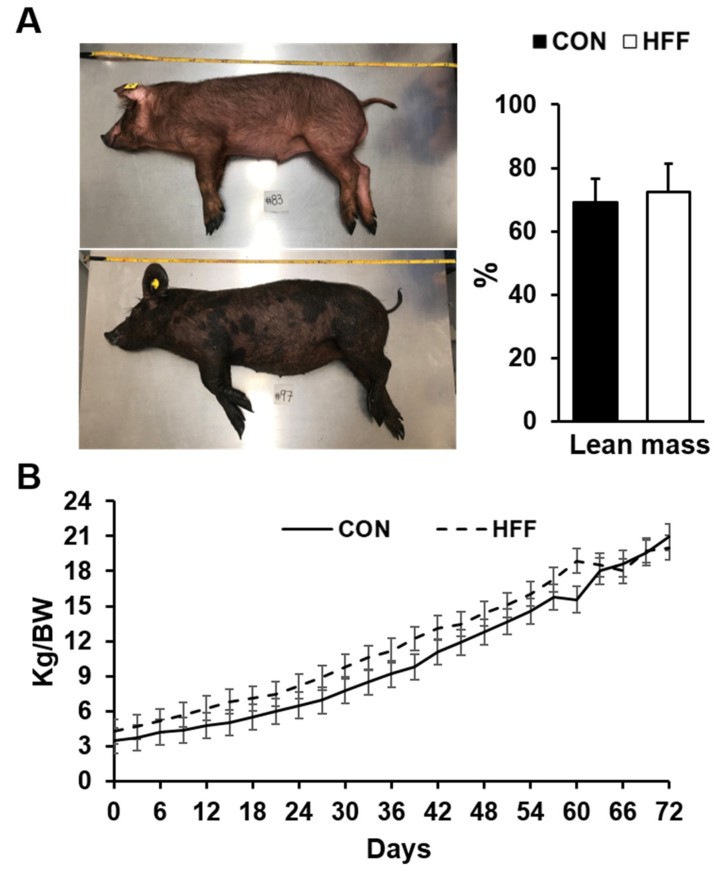
(**A**) Representative images of pigs on day 70 of the study. The percentage of lean mass composition did not differ between the control (CON) and high-fat, high-fructose (HFF)-fed pigs. Lean mass was calculated using the formula 100 × [8.588 + (0.465 × hot carcass weight) − (21.896 × 10th rib fat depth) + (3.005 × 10th rib loin muscle area)]/hot carcass weight. (**B**) Body weight did not differ between groups on day 70 of the study.

**Figure 4 biomedicines-11-03303-f004:**
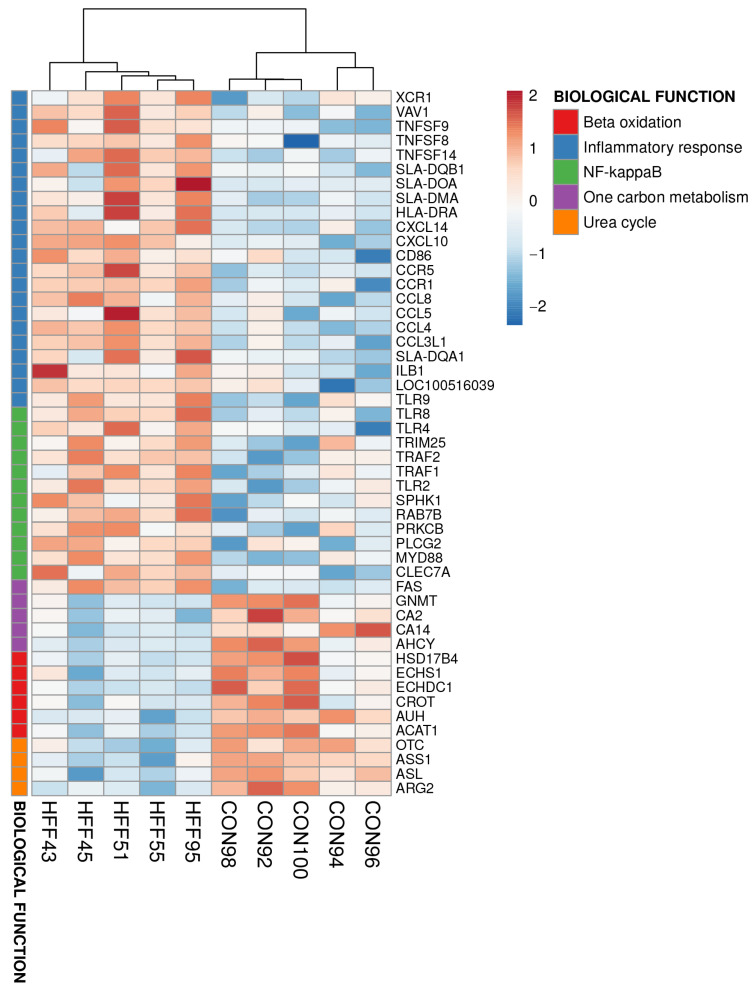
Heat maps of differentially expressed genes (5% FDR) between the HFF and CON groups in the liver of juvenile Iberian pigs on day 70 of the study. Columns are individual pigs, and rows are log-transformed read counts-per-million for each gene. Blue and red colors represent the row minimum and maximum values, respectively.

**Figure 5 biomedicines-11-03303-f005:**
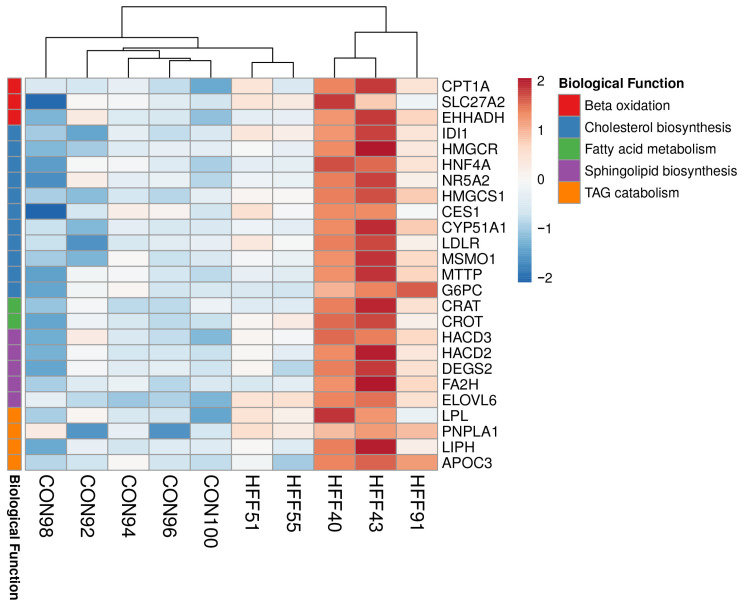
Heat maps of differentially expressed genes (5% FDR) between the HFF and CON groups in the distal ileum of juvenile Iberian pigs on day 70 of the study. Columns are individual pigs, and rows are log-transformed read counts-per-million for each gene. Blue and red colors represent the row minimum and maximum values, respectively.

**Figure 6 biomedicines-11-03303-f006:**
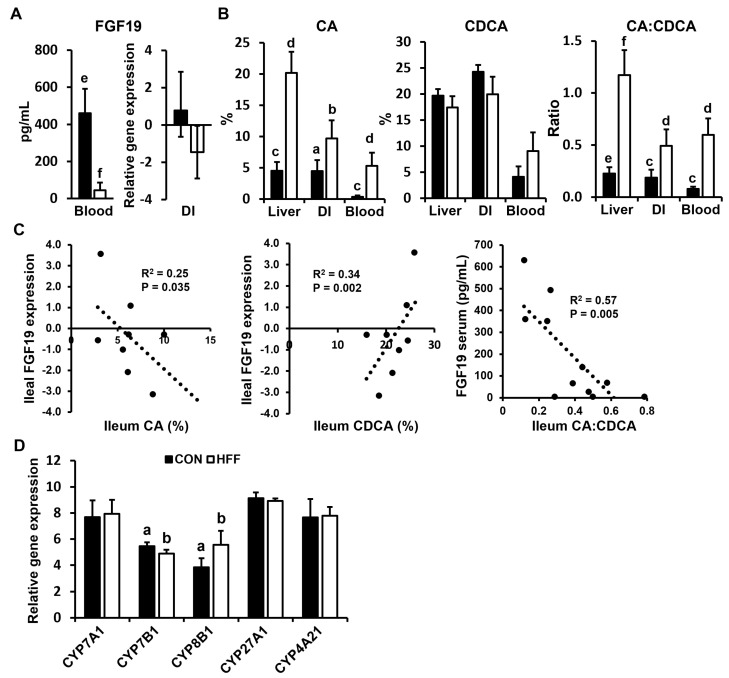
Quantification of fibroblast growth factor 19 (FGF19), cholic acid (CA), and chenodeoxycholic (CDCA) acid in the liver, distal ileum (DI), and peripheral blood of juvenile Iberian pigs using ELISA, metabolomics, and transcriptomics, respectively. Animals were fed either a control (CON) or a high-fat, high-fructose (HFF) diet for 10 consecutive weeks. (**A**) The HFF diet increased FGF19 levels in the blood and decreased FGF19 relative gene expression in the DI. (**B**) The HFF diet increased the percentage quantities of total CA species and the CA to CDCA ratio in the liver, DI, and blood. (**C**) Negative Pearson correlations between the percentage of total CA and CDCA and FGF19 expression in the DI and between the CA to CDCA ratio in the DI and FGF19 levels in the blood suggest a decrease in farnesoid receptor X (FXR) agonism in the bile acid pool. (**D**) The HFF diet increased hepatic expression of sterol 12-alpha-hydroxylase (CYP8B1), which increases the CA to CDCA ratio in the bile, and decreased sterol 7-alpha-hydroxylase (CYP7B1), which catalyzes the last step in the synthesis of CDCA. ^ab^
*p* ≤ 0.05, ^cd^
*p* ≤ 0.01, ^ef^
*p* ≤ 0.001.

**Table 1 biomedicines-11-03303-t001:** Ingredient composition of the control (CON) and high-fructose, high-fat (HFF) diets fed to juvenile Iberian pigs during 10 consecutive weeks. Values are expressed in percentages.

Item	CON	HFF
Whey protein concentrate ^1^	8.50	8.90
Fructose ^2^	0.00	12.0
Dextrose ^2^	6.00	3.00
Fat Pak 80 ^3^	3.20	0.00
Hydrogenated lard ^4^	0.00	3.70
Hydrogenated coconut oil ^2^	0.00	5.00
Corn oil ^5^	3.20	0.00
Xanthan gum ^6^	0.40	0.40
Vitamin premix ^7,8^	0.32	0.32
Mineral premix ^7,8^	1.20	1.20
Cholesterol ^7^	0.00	0.60
Water	77.2	64.4

^1^ 80% whey protein concentrate (Hilmar Ingredients, Hilmar, CA, USA). ^2^ Tate & Lyle, Hoffman Estates, IL, USA. ^3^ Advanced Fat-Pak 80 (Milk Specialties Global Animal Nutrition, Eden Prairie, MN, USA). ^4^ Armour, Grand Prairie, TX, USA. ^5^ Healthy Brand Oil Corporation, Queens, NY, USA. ^6^ NutraBlend, Neosho, MO, USA. ^7^ Dyets Inc., Bethlehem, PA, USA. ^8^ Provided per kilogram of vitamin premix (vitamin A: 4,409,171 IU; vitamin B-12: 15.4 mg; vitamin D-3: 661,376 IU; vitamin E: 17,637 IU; menadione: 1764 mg; D-pantothenic acid: 11,023 mg; riboflavin: 3307 mg; phytase: 200,000 FTU; niacin: 19,841 mg). Provided per kilogram of mineral premix (Cu: 11,000 mg; Fe: 110,000 mg; I: 200 mg; Mn: 26,400 mg; Se: 200 mg; Zn: 110,000 mg).

**Table 2 biomedicines-11-03303-t002:** Daily nutrient and metabolizable energy of the control (CON) and high-fructose, high-fat (HFF) diets fed to juvenile Iberian pigs during 10 consecutive weeks. Values are calculated and expressed as fed. BW, body weight.

Item	CON	HFF
Feed amount, L/kg BW/day	0.18	0.18
Dry matter, g/kg BW/day	40.8	62.6
Crude protein, g/kg BW/day	12.9	13.0
Metabolizable energy, kcal/kg BW/day	199.3	302.6
Carbohydrates, g/kg BW/day	12.8	29.1
Ether extract, g/kg BW/day	11.2	16.7
Amino acids, g/kg BW/day		
Arginine	0.30	0.29
Histidine	0.27	0.26
Isoleucine	0.78	0.76
Leucine	1.44	1.40
Lysine	1.11	1.07
Methionine	0.26	0.26
Cysteine	0.32	0.31
Phenylalanine	0.41	0.39
Tyrosine	0.32	0.32
Threonine	0.76	0.74
Tryptophan	0.20	0.20
Valine	0.69	0.69
Fatty acids, g/kg BW/day		
Caprylic	0.00	0.48
Capric	0.00	1.07
Lauric	0.00	3.39
Myristic	0.07	1.38
Palmitic	1.78	2.58
Stearic	0.67	1.28
Arachidic/eicosanoic	0.01	0.06
Behenic	0.00	0.00
Palmitoleic	0.14	0.23
Oleic	3.48	3.88
Gadoleic	0.06	0.08
Linoleic	3.86	0.96
Linolenic	0.04	0.06
Arachidonic	0.02	0.02
Cholesterol, g/kg BW/day	0.03	1.12
Calcium, g/kg BW/day	0.62	0.62
Phosphorus, g/kg BW/day	0.43	0.42
Fructose, g/kg BW/day	0.00	21.6
Dextrose, g/kg BW/day	10.8	5.40

**Table 3 biomedicines-11-03303-t003:** Serum hormones, glucose, and liver biochemistry in juvenile Iberian pigs fed either control (CON) or high-fat, high-fructose (HFF) diets for 10 consecutive weeks.

Item ^1,2,3^	CON	HFF
N° pigs (pen)	8 (4)	10 (5)
Sex (M/F)	6/2	6/4
Serum hormones and glucose		
Insulin, µIU/mL	9.38 ± 4.07	7.20 ± 2.03
Glucose, mg/dL	133.3 ^c^ ± 13.1	113.8 ^d^ ± 13.4
HOMA ^4^	2.67 ± 0.70	2.05 ± 0.69
Leptin, ng/mL	4.78 ± 1.75	4.52 ± 1.44
FGF19, pg/mL	405.6 ^e^ ± 143.6	74.7 ^f^ ± 74.5
Liver biochemistry		
ALT, U/L	19.5 ^a^ ± 3.48	62.18 ^b^ ± 40.37
AST, U/L	26.9 ^c^ ± 12.8	199.6 ^d^ ± 149.2
ALP, U/L	332.9 ± 59.8	391.6 ± 149.2
GGT, U/L	29.1 ^a^ ± 5.91	56.8 ^b^ ± 29.4
LDH, U/L	959.3 ^c^ ± 205.8	3114.2 ^d^ ± 1492.7
Total bilirubin, mg/dL	0.02 ^a^ ± 0.01	0.05 ^b^ ± 0.03
BUN, mg/dL	29.9 ^e^ ± 4.14	21.9 ^f^ ± 3.83
Albumin, g/dL	5.41 ± 0.43	5.60 ± 0.50
Total protein, g/dL	6.50 ± 0.67	6.83 ± 0.72
Creatinine, mg/dL	0.77 ^a^ ± 0.08	0.61 ^b^ ± 0.16
Lipid profile		
HDL, mg/dL	47.6 ^e^ ± 4.34	29.2 ^f^ ± 10.0
LDL, mg/dL	72.2 ^a^ ± 22.0	49.8 ^b^ ± 18.3
NEFA, mEq/L	0.25 ± 0.06	0.24 ± 0.14
Cholesterol, mg/dL	113.4 ^a^ ± 22.4	78.8 ^b^ ± 27.1
TAG, mg/dL	73.3 ± 16.0	72.3 ± 36.3

^1^ ALT, alanine aminotransferase; AST, aspartate aminotransferase; BUN, blood urea nitrogen; FGF19, fibroblast growth factor 19; GGT, gamma glutamyl transferase; HDL, high-density lipoprotein; IL1α, interleukin 1 alpha; LDH, lactate dehydrogenase; LDL, low-density lipoprotein; NEFAs, non-esterified fatty acids; TAG, triacylglyceride. ^2^ Data are presented as means ± SD. ^3^
*p*-values were calculated by a one-way ANOVA. Labeled means without a common letter differ: ^ab^
*p* ≤ 0.05, ^cd^
*p* ≤ 0.01, ^ef^
*p* ≤ 0.001. Values showing significant differences indicate alterations from the normal range for juvenile pigs. ^4^ HOMA: homeostatic model assessment values were calculated according to the formula [fasting insulin (µU/mL) × fasting glucose (mg/dL)]/405.

**Table 4 biomedicines-11-03303-t004:** Gene ontology terms up- and downregulated in the liver tissue of juvenile pigs fed a high-fat, high-fructose (HFF) diet compared with the control (CON). Functional enrichment analyses were performed on differentially expressed genes at a 5% false discovery rate to identify gene ontology (GO) terms pertaining to biological processes using the Database for Annotation, Visualization, and Integrated Discovery (DAVID) software version 6.8.

Biological Process	*p*-Value
Upregulated HFF vs. CON	
Immune response	≤0.001
Inflammatory response	≤0.001
NF-kappa B transcription factor activity	≤0.001
Neutrophil chemotaxis	≤0.001
Interferon-beta production	≤0.001
Positive regulation of ERK1 and ERK2 cascade	≤0.001
Positive regulation of interleukin-6 production	≤0.001
Antigen processing and presentation	≤0.001
Interleukin-8 production	≤0.001
Tumor necrosis factor production	≤0.001
Chemokine-mediated signaling pathway	≤0.001
Phagocytosis	≤0.001
Antigen processing and presentation by MHC class II	≤0.001
Toll-like receptor signaling pathway	≤0.001
Cytokine-mediated signaling pathway	≤0.001
Downregulated HFF vs. CON	
Fatty acid beta-oxidation	≤0.001
Urea cycle	≤0.001
One-carbon metabolism	≤0.01

**Table 5 biomedicines-11-03303-t005:** Gene ontology terms upregulated in the distal ileum of juvenile pigs fed a high-fat, high-fructose (HFF) diet compared with a control (CON). Functional enrichment analyses were performed on differentially expressed genes at a 5% false discovery rate to identify gene ontology (GO) terms pertaining to biological processes using the Database for Annotation, Visualization, and Integrated Discovery (DAVID) software version 6.8.

Biological Process	*p*-Value
Upregulated HFF vs. CON	
Sphingolipid biosynthetic process	≤0.001
Cholesterol biosynthetic process	≤0.001
Medium-chain fatty acid metabolic process	≤0.01
Fatty acid beta-oxidation	≤0.01
Cholesterol homeostasis	≤0.01
Triglyceride catabolic process	≤0.01

## Data Availability

Datasets cannot be publicly shared due to privacy issues. They can be provided by email upon request.

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
