# Peer review of "Decreased FXR Agonism in the Bile Acid Pool Is Associated with Impaired FXR Signaling in a Pig Model of Pediatric NAFLD"

_biomedicines, 2023, doi:10.3390/biomedicines11123303_

Round 1
Reviewer 1 Report
Comments and Suggestions for Authors
fatty liver disease. This is a well-written paper containing interesting results which merit publication. For the benefit of the reader, however, a number of points need clarifying and certain statements require further justification. These are given below.
1) Please use same format in table 4 and table 5.
2) Little discussion of the significantly different results shown in Table 4 and Table 5. Please add them to the discussion.
3) In discussion, authors described about expression of CYP8B1 are upregulated by the PPAPgamma. How about other fat maturating factor such as Fasn, Perilipin, GLUT4 C/EBPalpha?
4) The body weight was not a significant difference between control and HFF-fed piglet.
As the author stated, fatty liver in children is often attributed to obesity. What makes you think that this is a significant difference from this piglet model, and why do you think it is a model for the disease? Please add an explanation for the reader.
5) Figure 1 has many abbreviations, please add an explanation of each to the legend.
6) The authors used the rectum for their analysis. Since the animals are euthanized, it should be possible to consider the small intestine or other sites.
Reviewer 2 Report
Comments and Suggestions for Authors
Concerning the manuscript;
Ref.:Ms. biomedicines-2669893
Impaired FGF19 signaling in NAFLD pigs is associated with decreased FXR agonism of bile acid pool
The paper presented for the review is written well, the selection of references is correct, but several issues should have included or discussed before the publication of your manuscript;
1- The numbers of animals in each group should be clearly presented at the Animals and Experimental Design section.
2- Blood samples were centrifuged at 2,100 rpm for 15 min at 4ËšC, should be corrected to relative centrifugal force (RCF).
3- Serum and plasma were stored at -80ËšC; the authors separate and use serum or plasma or both, clarify this point.
4- The authors need to discuss all data in their study in the discussion section.
5- As displayed in the result section glucose, creatinine, blood urea nitrogen, cholesterol and high- and low-density lipoproteins were decreased in HFF compared with CON group , explain and discuss.
6- Figure 2A, should add the magnification power for histological images.
Reviewer 3 Report
Comments and Suggestions for Authors
Dear authors
The study is interesting, it aims to determine whether an alteration of FXR-FGF19 could be associated with fatty liver in pigs. However, I believe that this objective has not been met. I detail the points that need to be modified in the article for a better understanding.
(1)The title should indicate that the study is aimed at humans.
(2)Lines 1 and 2 of the introduction: there may be abundant cases of paediatric fatty liver disease, improve the wording. Should mention that one of the most important causes of fatty liver in the US is fatty liver due to food or medication, or the main cause of fatty liver. 0.7% is too low a value to say that it is the leading cause, please revise and reword.
(3)In the methodology, add the literature reference that a high fructose and high-fat diet is a standard way to produce fatty liver.
(4)Upload in supplementary material the permission for the use of experimental animals.
(4)In table 3 (blood profile) of results, indicate whether values showing significant differences are within the normal clinical range or are values that would indicate pathological alterations.
(5)In Tables 4 and 5, the genes involved in each signalling pathway mentioned should be included. And in supplementary material the primers that were used. It is very general to just put the biological process, check.
(6)The text describing Figure 4 should contain the description and the meaning of having performed r2 (its significance).
(7)Neither the results nor the discussion indicates how FXR-FGF19 is altered, saying that it only increased/decreased its activity or expression only indicates a situation at that time. What would be indicated would be to see if and to what extent its receptors or a canonical signalling pathway are altered. Revise and refocus the discussion.
(8)Suggest changing the title to be more in line with the results described.
Regards...
Round 2
Reviewer 1 Report
Comments and Suggestions for Authors
There are no comments.
Reviewer 2 Report
Comments and Suggestions for Authors
The authors edited all inquiries, I have no further comments.
Reviewer 3 Report
Comments and Suggestions for Authors
Dear Author
After reviewing the manuscript, I have observed that the authors have collected observations based on my suggestions, which is why I consider that the work meets the conditions for acceptance.